# The Effect of Various Poly (*N*-vinylpyrrolidone) (PVP) Polymers on the Crystallization of Flutamide

**DOI:** 10.3390/ph15080971

**Published:** 2022-08-06

**Authors:** Dawid Heczko, Barbara Hachuła, Paulina Maksym, Kamil Kamiński, Andrzej Zięba, Luiza Orszulak, Marian Paluch, Ewa Kamińska

**Affiliations:** 1Department of Pharmacognosy and Phytochemistry, Faculty of Pharmaceutical Sciences in Sosnowiec, Medical University of Silesia in Katowice, 41-200 Sosnowiec, Poland; 2Institute of Chemistry, Faculty of Science and Technology, University of Silesia in Katowice, 40-007 Katowice, Poland; 3Institute of Material Science, Faculty of Science and Technology, University of Silesia in Katowice, 41-500 Chorzów, Poland; 4Institute of Physics, Faculty of Science and Technology, University of Silesia in Katowice, 41-500 Chorzów, Poland; 5Department of Organic Chemistry, Faculty of Pharmaceutical Sciences in Sosnowiec, Medical University of Silesia in Katowice, 41-200 Sosnowiec, Poland

**Keywords:** flutamide, poly(*N*-vinylpyrrolidone), polymer tacticity, activation barrier of crystallization, broadband dielectric spectroscopy

## Abstract

In this study, several experimental techniques were applied to probe thermal properties, molecular dynamics, crystallization kinetics and intermolecular interactions in binary mixtures (BMs) composed of flutamide (FL) and various poly(*N*-vinylpyrrolidone) (PVP) polymers, including a commercial product and, importantly, samples obtained from high-pressure syntheses, which differ in microstructure (defined by the tacticity of the macromolecule) from the commercial PVP. Differential Scanning Calorimetry (DSC) studies revealed a particularly large difference between the glass transition temperature (*T*_g_) of FL+PVPsynth. mixtures with 10 and 30 wt% of the excipient. In the case of the FL+PVPcomm. system, this effect was significantly lower. Such unexpected findings for the former mixtures were strictly connected to the variation of the microstructure of the polymer. Moreover, combined DSC and dielectric measurements showed that the onset of FL crystallization is significantly suppressed in the BM composed of the synthesized polymers. Further non-isothermal DSC investigations carried out on various FL+10 wt% PVP mixtures revealed a slowing down of FL crystallization in all FL-based systems (the best inhibitor of this process was PVP M_n_ = 190 kg/mol). Our research indicated a significant contribution of the microstructure of the polymer on the physical stability of the pharmaceutical—an issue completely overlooked in the literature.

## 1. Introduction

Polymers are long-chain organic substances consisting essentially of repeated chemical units (called “mers”) linked together [1]. Based on origin, one can distinguish three main classes of these compounds: natural (e.g., cellulose and collagen), semisynthetic (e.g., (hydroxypropyl) methylcellulose (HPMC)) and synthetic (poly(*N*-vinylpyrrolidone) (PVP), poly(ethylene glycol) (PEG) and polystyrene (PS)) [2,3]. Polymers from each group play essential and ubiquitous roles in everyday life [4]. Moreover, some of them (exclusively non-toxic, biocompatible and physiologically inert macromolecules, accepted by Food and Drug Administration (FDA) and European Medicines Agency (EMA)), have application in pharmaceutical and biomedical industries, e.g., for drug protection, taste masking, targeted delivery or the controlled release of a given active substance, and also as tablet binders, solubilizers, emulsifiers, film coating agents (in solid dosage forms) and so on [1,2,3]. Among them, there are, e.g., vinyl lactam-based (co)polymers (i.e., PVP [5,6,7] and its copolymer with vinyl acetate (PVP/VA) [8], Kollidon^®^VA64/Plasdone^TM^S-630 [9,10]; poly(vinyl caprolactam)-poly(vinyl acetate)-poly(ethylene glycol) graft copolymer (Soluplus) [7,11]); poly(methacrylate)s derivatives (Eudragit^®^ L100-55, L100, S-100 and EPO grades) [5,12,13]; cellulose derivatives (HPMC [5,14,15], (hydroxypropyl) methyl cellulose acetate succinate (HPMCAS) [5,14]); and poly(ethylene glycol)s (PEG) [16,17,18]), which are used for the preparation (generally using methods based on solvent evaporation or melt cooling [19]) of amorphous solid dispersions (ASDs)/binary mixtures (BMs) with poorly water-soluble active pharmaceutical ingredients (APIs) from II and IV groups of Biopharmaceutical Classification System (BCS) [20,21].

Importantly, such a procedure results in improving apparent solubility/dissolution rates and, consequently, the bioavailability of drug substances (due to the fact that APIs in the amorphous form have no long-range crystallographic order and are characterized by higher free energy and greater chemical and thermodynamic activity as compared to the crystalline substances [22]). It is worth emphasizing that a given polymer that is physically and chemically stable, acts as an inactive stabilizer (a crystallization inhibitor) for disordered/labile pharmaceuticals in the ASD [14,23]. The physicochemical stabilization with its use results mainly from increasing the glass transition temperature, *T*_g_ (and consequently reducing molecular mobility of the system), as well as from strengthening intermolecular interactions between API and macromolecule (causing inhibition of nucleation and crystal growth) [24,25]. However, it should be added that this finding may also be affected by other factors, such as configurational entropy and enthalpy, Gibbs-free energy, humidity, mechanical stress and preparation methods (and conditions, e.g., cooling rate) [26]. Herein, one can also mention the volume fraction effect, which is important in the case of hybrid polymer-based nanocomposites. In such systems, the suppressed crystallization of the polymers, as well as self-assembly of the nanoparticles, has been discussed as related to this phenomenon [27,28,29,30]. Among FDA (or EMA)-approved medical preparations that are based on ASDs, one can list Modigraf^®^ (Tacrolimus/HPMC), Novir^®^ (Ritonavir PVP-VA), Cesamet^TM^ (Nabilone-PVP), Cymbalta^®^ (Duloxetine/HPMCAS) or Gris-PEG™ (Griseofluvin/PEG) [23].

Besides the beneficial properties of the polymers mentioned above, there are essential problems connected with the development of ASDs such as (i) the poor solubility of the drug in the polymer matrix (which results in the incorporation of a larger amount of excipient and, hence, increasing the final dosage form volume) and the (ii) high hygroscopicity of the macromolecule (which affects the molecular mobility of the active substance) [31,32]. Moreover, due to the high molar-mass distribution (i.e., high dispersity) in the case of several commercially available macromolecules (e.g., PVP—a water-soluble and biocompatible) or different degrees of the substitution of functional groups (cellulose derivatives, e.g., HPMC), there is a risk of phase microseparation in ASDs, resulting in the recrystallization of the incorporated amorphous pharmaceutical. In this context, it seems reasonable to synthesize and use polymers with strictly controlled parameters (i.e., low/moderate dispersity (Đ), tailored molecular weight (M_n_), high functionality, chain-end fidelity, etc.) to obtain homogeneous, stable solid dispersions with desirable pharmacokinetic properties.

In this paper, we present the results of calorimetric, dielectric and infrared studies carried out on ASDs composed of flutamide, FL (an anticancer drug for prostatic carcinoma, belonging to BCS class II), with linear PVPs. Basically, three different PVPs (that is (i) commercially supplied PVP K90 (PVPcomm.) obtained in an aqueous solution, (ii) self-synthesized PVP via high-pressure free-radical polymerizations (HP-FRP) in bulk—a novel synthetic methodology, which enables the preparation of well-defined PVP of high purity using the “*greenest*” synthetic strategy [33,34], and (iii) self-synthesized hydroxyl-terminated PVP (PVP-OH) via HP CT-FRP mediated by 2-isopropoxyethanol, acting simultaneously as a solvent and chain-transfer agent, CT) have been the subject of our research. The full characteristic of these macromolecules, including ^1^H, ^13^C NMR and SEC analyses, is presented in the Appendix A. In particular, in our studies, we focus on determining the impact of these excipients, with defined macro- and micro-(structural) parameters, on the progress and activation barrier for crystallization, as well as characterizing intermolecular interactions in the examined BMs. 

It should be mentioned that FL has been previously investigated in amorphous binary systems with polymers such as Kollidon^®^VA64 [35,36], poly(vinyl acetate) (PVAc) [37], PVP K90 (a commercial sample, produced at 0.1 MPa, for which the number average molecular weight (M_n_) and Đ are poorly controlled), HPMC, Eudragit EPO, PEG 8000 [38] and in ternary mixtures with bicalutamide and poly(methyl methacrylate-co-ethyl acrylate) (MMA/PEA) or PVP K90 [39]. The subject of these studies was not only the influence of a given macromolecule on the physical stability/crystallization tendency but also a determination of the solubility of FL in the selected polymer matrices (with the use of broadband dielectric spectroscopy, both at ambient and high-pressure conditions), as well as studying the interactions between components of BMs and the effect of various polymers on FL precipitation. Herein, for the first time, various PVPs (including a hydroxyl-terminated derivative of this polymer, PVP-OH) with tailored M_n_ (=190 kg/mol or 90 kg/mol) and strictly controlled parameters (dispersity and chain-end fidelity) were applied to form ASDs with FL. Note that, previously, such macromolecules (with linear and star-shaped topologies [33,34]), produced via solvent-free HP strategy, have been used by some of us to form micellar drug delivery systems with metronidazole (MTZ) and to examine how these matrices affect the amorphization of API, its encapsulation, the stability of MTZ-loaded micellar structures and their in vivo release from the carrier [40]. 

## 2. Results and Discussion

As a first, we have carried out calorimetric measurements on neat FL, various PVPs (including PVP-OH derivative)—see Figure 1 and their mixtures (with 10 and 30 wt% of the second component). It should be noted that we did not examine binary systems with a higher content of PVP due to miscibility limitations (see the Materials and Methods section). DSC curves obtained upon heating the glassy samples are presented in Figure 2 (panels a–d).

As illustrated, the thermogram of each polymer (see the insets) reveals a single thermal event corresponding to the glass transition at *T*_g_ close to 450 K. In turn, in the case of neat FL and its ASDs with different PVP/PVP-OH content, besides the liquid-glass transition at lower *T*, there are additional peaks related to cold crystallization (exothermic process) and melting (endothermic process), respectively. The exception is FL+30 wt% PVP (M_n_ = 190 kg/mol) and FL+PVP (M_n_ = 90 kg/mol) systems, where such additional thermal events are not detected. Hence, it can be concluded that these two new synthetized PVPs (with controlled M_n_ and Ð), in contrast to PVPcomm., inhibit FL crystallization from the amorphous state when its content in the mixture is sufficiently high (30 weight percent). In the context of PVPcomm. (M_w_ = 340 kg/mol, M_n_ = 109 kg/mol), it is worth mentioning that the same polymer (available in the market) but with lower M_w_ (=58 kg/mol) was an effective stabilizer of amorphous FL at nearly the same concentration (strictly 29 wt%) of the polymer [37]. One can suppose that the slow down of the crystallization in the case of the two examined systems herein (>30 wt% of PVPsynt. M_n_ = 90 and 190 kg/mol) can be related to the greater volume fraction of polymers. Importantly, a similar phenomenon was argued to be responsible for the suppression of the polymer crystallization and self-assembly process of nanoparticles in polymer-based nanocomposite materials [28,29,30].

The values of *T*_g_ determined from DSC studies for examined BMs and neat FL are presented in panel (e) of Figure 2. It is well-observed that the glass transition temperature increases with the increasing content of the polymer in each mixture. Interestingly, the difference between *T*_g_^DSC^ of FL+10 wt% and FL+30 wt% of PVPcomm. systems is negligible (Δ*T*_g_^DSC^ = 1 K). On the other hand, the more significant difference (Δ*T*_g_^DSC^ = 20–30 K) occurs for the other mixtures. This is a quite intriguing finding that cannot be explained considering only variations in Ð or M_n_ between commercial and synthesized PVPs. There must be other important factors that are responsible for such a peculiar change of the *T*_g_ between considered solid dispersions. 

To explain this peculiarity, one needs to realize that the *T*_g_ of PVP is quite sensitive to the presence of unreacted monomer, moisture, and the implemented preparation strategy (character of initiating species, bulk vs. solvent polymerization). Interestingly, by manipulating all these parameters, we can directly affect the PVP molecular weight, dispersity, cross-linking degree and branching caused by chain-transfer processes [41]. However, considering that the *T*_g_s of all recovered polymers were close to each other and collected NMR spectra revealed their high purity (lack of additional signals coming from unreacted monomer, water, solvent, etc.), the above-mentioned argumentation must be rejected. The other important factor that may influence the evolution of the *T*_g_ in BMs is stereoregularity (syndio, iso- or atactic arrangement of the lactam moiety along the main chain) of the produced PVPs. In this context, it is worth stressing that, currently, it is well-established that a polymer microstructure can depend on both “internal” (e.g., solvents/additives polarity/density) and “external” factors (e.g., given values of high pressure/temperature). For example, PVPs produced via thermally induced radical polymerization (*p* = 0.1 MPa) are mostly atactic [42]. In the vinyl pyrrolidone (VP) polymerization conducted in the presence of fluoroalcohols/anionic surfactants, some increase in syndiotacticity can be observed [43]. In turn, the implementation of Lewis acid catalysts to VP FRP increases the PVP’s isotacticity [44]. These phenomena are mostly related to electron pair donor–acceptor interactions between VP and given additives. Importantly, even though the stereoregularity of PVP seems to be a very important parameter describing its microstructure. The above issue is barely investigated and almost overlooked in the literature, especially in the cases in which this polymer is applied in pharmaceutical formulations of disordered APIs. The reason of such situation is the fact that there is only one leading supplier of PVP on the market. Hence, the microstructure of delivered macromolecules is assumed to be invariant since the same synthesis strategy is adopted for their production. The situation becomes less clear when we compare PVPs obtained via different methods, such as herein by applying high pressure. Therefore, to describe the microstructure of the polymers used to prepare binary mixtures, carbon-^13^C nuclear magnetic resonance spectra (^13^C NMR in D_2_O) of the investigated commercial and synthesized PVPs [45,46] (see the Appendix A) were analyzed in detail. Notably, PVPs (M_n_ = 190 kg/mol and 90 kg/mol) and PVP-OH samples produced by us (via HP-FRP and HP CT-mediated FRP, respectively) revealed ultra-high purity (lack of moisture and uncreated monomer), and the only difference between them was the presence or absence of the solvent during their preparation. The obtained results (analysis of triad and tetrad sequences) are presented in Figure 3 and Appendix A. Let us take a look firstly at the microstructure of PVPcomm. determined by the ^13^C NMR carbonyl region (see Figure 3). As illustrated, the PVPcomm. (PVP K90) sample is dominated by atactic fractions, with a relatively high content of isotactic ones, and this result is in line with other findings in the literature [47]. In turn, samples synthesized herein, i.e., PVP with M_n_ = 190 and 90 kg/mol (produced via solvent-free HP-FRP, 250 MPa) showed a significant increase in isotactic fractions, i.e., 15–16%. Similar conclusions were derived from the analysis of tetrad sequences determined from *β*-methylene protons (^13^C NMR, D_2_O, please see Appendix A). The above effect (i.e., an increase in the number of isotactic units) was previously reported for HP-FRP of methyl methacrylate [48] or sterically congested α-acrylates [49]. Finally, a short note should be added to PVP-OH prepared via thermally initiated HP (*p* = 250 MPa) reactions, in which an additional alcoholic chain transfer agent (2-isopropoxyethanol) was used. Interestingly, when analyzing this sample, we noticed that the number of isotactic fractions is slightly lower than those noted for PVPs synthesized via solvent-free HP-FRP (M_n_ = 190 kg/mol and 90 kg/mol), but at the same time, it was significantly higher than that observed in PVPcomm. It seems clear to us that the simultaneous action of solvent and high-pressure was responsible for this result. Without mentioning specific details, it should be summarized that gaining control over polymer tacticity is a reasonably complex aspect influenced by many factors.

Interestingly, the mentioned differences in the tacticity do not significantly affect the glass transition temperature of the studied polymers (they are characterized by a similar *T*_g_, in the range 449–452 K). Hence, obtained results stay in contrast to the literature data showing that there are normally much greater variations in the *T*_g_ of more or less polar macromolecules (e.g., poly(methyl methacrylate) (PMMA), poly(2-methoxyethylacrylate) (PMEA) [50,51,52,53], polystyrene [54], etc. [55]), differing in tacticity (i.e., with various content of isotactic (i), syndiotactic (s) and atactic (a) fractions in the sample). Moreover, in the case of polypropylene, a variation in the microstructure leads to a change in the degree of crystallinity and mechanical properties [55]. The observed discrepancies might be related to the highly polar character of PVP and, contrary to the other polymers, its highly hydrophilic nature. One can also suppose that the residual water, which is always present in the structure of this polymer (it is very difficult to remove it even by heating/annealing), can weakly interact via H-bonds with the molecules of macromolecule. For this reason, the effect of *T*_g_ of PVP on tacticity is clearly lower when compared to other mentioned polymers.

On the other hand, to the best of our knowledge, the correlation between the tacticity of a given polymer and the *T*_g_ of the BM formed by low molecular weight substance (including API) and the macromolecule has not been investigated so far. There are only reports on blends composed of PMMA with various tacticities, i.e., a-PMMA/s-PMMA, i-PMMA/s-PMMA, as well as a-PMMA/i-PMMA [56,57]. It should be noted that in these papers, the analysis of the *T*_g_ of the binary system versus s-PMMA or a-PMMA (wt%) in the mixtures revealed a marked increase in the glass transition temperature (about 85 K in the two latter cases and 17 K in a-PMMA/s-PMMA mixture) with increasing contents of a given conformer in the blend. One can suppose that the results determined for FL+PVPcomm. and FL+other PVPs mixtures (i.e., small or large variations, respectively, between *T*_g_ of the systems with 10 and 30 wt% of the excipient) may be explained considering differences in the tacticity (the content of respective fractions) of the polymer forming ASDs.

After calorimetric studies, we have carried out dielectric spectroscopy measurements to characterize the molecular dynamics of FL and FL+10% wt. PVP (PVP-OH) systems and checked at which *T* the crystallization process occurs. Note that due to the fact that there were no traces of crystallization in thermograms registered for all considered FL + 30 wt% polymer mixtures, such experiments were not performed on those samples. The results of BDS measurements (*ε*′′ vs. frequency (*f*) dependencies) carried out in a wide temperature range are presented in panels a–e in Figure 4. As illustrated, except for the DC-conductivity (DC), the spectra registered at *T* > *T*_g_ exhibit one well-resolved loss peak corresponding to the structural (*α*)-relaxation, which moves towards higher *f* with increasing *T*. In Figure 5a, we compared the shape of *α*-process for FL and all examined mixtures at the indicated *T* close to *T*_g_ (the maxima of the peaks near 1 Hz) by fitting the presented data by means of the one-sided Fourier transform of the Kohlrausch–Williams–Watts (KWW) function [58,59] (solid lines). It turned out that the presence of a small amount of each polymer results in a broadening of *α*-dispersion (compared to neat FL, for which the stretched exponent, *β*_KWW_ = 0.86). The lowest values of this parameter (which means the broadest structural peak) were obtained for BMs with two synthesized PVP: M_n_ = 190 kg/mol and 90 kg/mol (0.63 and 0.65, respectively). Slightly higher *β*_KWW_ (~0.70) was determined for two other mixtures, i.e., with PVPcomm. and PVP-OH. 

We have also analyzed the spectra of each system presented in Figure 4 with the use of the Havriliak–Negami function [60]:(1)$$\varepsilon^{\prime \prime }\;\left(\omega \right)=\frac{\sigma_{DC}}{\varepsilon_{0}\omega }+\varepsilon_{\infty }+\frac{∆\varepsilon }{{\left[1+{\left(i\omega \tau_{HN}\right)}^{a}\right]}^{b}}$$
where *ε*_0_ is the vacuum permittivity, $\sigma_{DC}$ is the dc-conductivity, *ω* (=2π*f*) is the angular frequency, $\varepsilon_{\infty }$ is the high-frequency limit permittivity, ∆ε is the dielectric strength, *τ_HN_* is the HN relaxation time and *a* and *b* represent the symmetric and asymmetric broadening of the given relaxation peak. Then, based on the fitting parameters determined above, structural relaxation times (*τ_α_*) were calculated from the following formula [60]:(2)$$\tau_{\alpha }=\tau_{HN}{\left[sin\frac{a\pi }{2+2b}\right]}^{-1/a}{\left[sin\frac{\mathrm{ab}\pi }{2+2b}\right]}^{1/a}$$
and next plotted as a function of 1000/*T* in Figure 5b. We further fitted these dependencies by the Vogel–Fulcher –Tammann function: (3)$$\tau_{\alpha }=\tau_{VFT}exp\left(\frac{D_{T}T_{0}}{T-T_{0}}\right),$$
where $\tau_{VFT}$ is the relaxation time at finite temperature and $D_{T}$ is the strength parameter or fragility, whereas $T_{0}$ represents *T* at which $\tau_{\alpha }$ tends to infinity, we determined *T*_g_ (here defined as a *T* at which *τ_α_* = 100 s) for FL and each FL+ 10 wt% polymer system. As observed in Figure 5b and Figure 2e, as well as in Table 1, there is a good correspondence between *T*_g_^BDS^ and *T*_g_^DSC^ (heating rate 10 K/min); Δ*T*_g_ = ±3 K. Moreover, the values of this parameter for FL+PVP M_n_ = 190 kg/mol and FL+PVP M_n_ = 90 kg/mol systems are slightly higher than the *T*_g_ of FL+PVP-OH and FL+PVPcomm. mixtures.

Importantly, from Figure 4, it can be observed that the onset of the crystallization process in FL+10 wt% PVPcomm., reflected in the lowering amplitude of the *α*-process, occurs at 297 K, which is barely 2 K higher than *T*_onset_ in a neat FL system. On the other hand, for the three other BMs (containing synthesized polymers: PVP M_n_ = 90 kg/mol, PVP M_n_ = 190 kg/mol and PVP-OH), the crystallization of API begins at higher *T*: 311, 307 and 301 K, respectively). It is well-visualized in panel (f) of Figure 4, presenting the difference between *T*_onset_ and *T*_g_ for FL and all considered mixtures.

To verify how a given polymer influences the progress and activation barrier of FL crystallization in the examined systems, we carried out non-isothermal calorimetric measurements. DSC curves measured with various heating rates (ϕ) for neat FL and FL+10 wt% PVP M_n_ = 190 kg/mol system are shown in panels (a) and (b) of Figure 6. Analogical data for three other BMs are presented in Appendix A. It is well-observed that there are two or three endothermic processes in the thermograms collected at 5, 10, 20 and 30 K/min heating rates (ϕ). The first one, which emerges at lower *T*, is associated with the glass-transition event. In turn, the other ones (one in the case of neat FL and its binary system with PVPcomm., PVP-OH; two in the case of FL+PVP M_n_ = 190 kg/mol and FL+PVP M_n_ = 90 kg/mol mixtures) are related to the melting process. Moreover, in all samples, for each ϕ, the presence of an exothermic event indicating the crystallization of FL can be observed.

Based on the results of non-isothermal crystallization experiments carried out at various ϕ, the relative degree of crystallinity (X) as a function of *T* has been obtained:(4)$$X\left(T\right)=\int_{T_{0}}^{T}\left(dH_{c}/dT\right)dT/\int_{T_{0}}^{T_{\infty }}\left(dH_{c}/dT\right)dT,$$
where $dH_{c}$ represents the enthalpy of crystallization released during an infinitesimal temperature interval and $T_{0}$ denotes the initial crystallization temperature, while T and $T_{\infty }$ denote the crystallization temperature at time t and the temperature at the end of the crystallization process. Representative data (*X* vs. *T* dependence) for neat FL are provided in Appendix A.

Having $X\left(T\right)$ for each system, we were able to determine $X\left(t\right)$ by transforming the temperature axis to the time (*t*) axis using the following formula [61].
(5)$$t=\left(T-T_{0}\right)/\phi$$

The plots of a relative degree of crystallinity against crystallization time t at different ϕ for neat FL and FL+10 wt% PVP M_n_ = 190 kg/mol system are shown in panels (a) and (b) of Figure 7, while analogical data for other BMs are presented in Appendix A. As observed, in each case, the crystallization slows down with decreasing heating rates. Moreover, the curves determined for each binary system at lower ϕ (5 and 10 K/min) seem to have a different shape than those of neat FL registered at the same ϕ. This might suggest the two-stage crystallization process. To determine constant rates of non-isothermal crystallization in a neat API (FL) system, the Avrami equation was applied [62,63]:(6)$$X\left(t\right)=1-\exp\left(-{\left(k_{1}t\right)}^{n}\right)$$
where $k_{1}$ is a rate constant, and n is the Avrami exponent, which depends on the crystal morphology and crystallization mechanism [64].

In turn, for FL+10 wt% PVP/PVP-OH mixtures, due to the inability to obtain a good fit with the use of Equation (6), we have used the following expression:
(7)$$X\left(t\right)=\left(1-x\;\right)\exp\left(-{\left(k_{1}t\right)}^{n}\right)-\left(1-x\right)e\mathrm{xp}\left(-{\left(k_{2}t\right)}^{m}\right)$$
which is a modification of the Avrami model for considering two-stage crystallization (herein named “double Avrami” model). x in Equation (7) parametrizes the relative contributions of each crystallization step and $k_{1}$ and $k_{2}$ are rate constants, while n and m are fit parameters. It is worth pointing out that according to the literature reports on a two-stage crystallization process (however taking place from solution/fluid, not from the solid state), in the first step, dense liquid (disorder) clusters are formed, while in the second step, the molecules in the mentioned clusters rearrange their orientations to form ordered nuclei (crystals). Such a mechanism, which was suggested also in the case of, e.g., triblock Janus assemblies [65,66,67], proteins [68], colloids [69] and biomaterials [70], is sometimes more favorable than the one-step formation of crystalline nuclei. However, it should be noted that since we measured solid-state systems, the two-stage process might be due to the change of one polymorphic form into another during the course of crystallization.

In panels (c) and (d) of Figure 7, dependences of log_10_$k_{1}$ and log_10_$k_{2}$ versus ϕ are presented. As illustrated (Figure 7c, Table 1), a small amount of each polymer (10 wt%) causes the slow down of FL crystallization. It is particularly noticeable at higher heating rates: 20 and 30 K/min (smaller $k_{1}$ when compared to that of neat FL). PVP M_n_ = 190 kg/mol is the excipient for which its impact on such behavior is the most significant (the lowest values of $k_{1}$ at each ϕ; see Table 1). Comparing polymers with the same/nearly the same M_n_, it can be stated that PVP M_n_ = 90 kg/mol and PVP M_n_ = 190 kg/mol, compared to PVPcomm. and PVP-OH, respectively, suppress/slow down the API crystallization a little bit more (Figure 7c). In the context of the constant rate $k_{2},$ one can add that its value determined for all binary systems at a given ϕ was comparable (Table 1). It should also be mentioned that the values of the exponent n obtained from the analysis of the data for neat FL using Equation (6) and FL+10 wt% PVP (PVP-OH) using Equation (7) changed generally in the range of 2–3—a somewhat lower *n* (1.3–1.7) was determined only for neat FL at lower ϕ, see Table 1). In turn, the values of the parameter m (Equation (7)) were clearly higher and unreliable. Therefore, we will not present (in Table 1) and discuss them.

Subsequently, we have also calculated the activation energy of non-isothermal crystallization ($E_{cr}$) for the examined FL+10 wt% polymer (various PVPs) BMs. For this purpose, the thermograms presented in Figure 6 and Appendix A were analyzed with the use of two approaches, proposed, respectively, by Augis and Bennett [71] as well as Kissinger [72]. The former method considers the onset temperature of crystallization ($T_{o}$).
(8)$$ln\left(\frac{\phi }{T_{p}-T_{o}}\right)=C_{AG}-\frac{E_{cr}}{RT_{p}}$$

In turn, the latter one is based on the variation of the crystallization peak temperature ($T_{p}$) with ϕ.
(9)$$ln\left(\frac{\phi }{T_{p}^{2}}\right)=C_{K}-\frac{E_{cr}}{RT_{p}}$$

$C_{AG}$ and $C_{K}$ in the above expressions (Equations (8) and (9)) are fitting parameters, while R is a gas constant. The obtained results (together with *E*_cr_ determined for each system) are presented in Figure 8. It should be noted that the values of $T_{p}$ and $T_{o}$ are provided in Table 1. As observed in Figure 8c, the Kissinger methods yielded a higher *E*_cr_ than the Augis–Bennett approach. Similar conclusions were derived from the analysis of the crystallization data obtained for other APIs/BMs, e.g., salol [73] or naproxen-acetylated saccharide systems [74]. Generally, the activation barrier for API crystallization in a system with 10 wt% of PVP comm. is relatively high and slightly smaller than that obtained for neat FL (Augis-Bennett:~150 kJ/mol, Kissinger~200 kJ/mol). For other mixtures, somewhat lower, comparable values of *E*_cr_ (Augis-Bennett: from 119–138 kJ/mol; Kissinger: from 167–182 kJ/mol) were determined. Based on the above, it can be concluded that the presence of each of the examined polymers (both commercial and synthesized samples) only slightly affects the *E*_cr_. In the context of neat FL, it should be mentioned that the obtained *E*_cr_ is significantly greater when compared to the values calculated using the Augis–Bennett and/or Kissinger approaches for other APIs, e.g., sildenafil (*E*_cr_~100–120 kJ/mol) [75], carbamazepine (*E*_cr_~100 kJ/mol for *ϕ* > 7.5 K/min) [76], salol (*E*_cr_~80–90 kJ/mol) [73] or biclotymol (*E*_cr_~50 kJ/mol) [77]. One can mention ketoprofen, for which an even higher *E*_cr_ from the Kissinger method (~264 kJ/mol) with respect to that obtained for FL was determined [78].

Aside from the molecular dynamics studies and crystallization kinetics, we also applied ATR-FTIR spectroscopy to get insight into intermolecular interactions occurring between FL molecules and PVP/PVP-OH polymers. The representative spectra of neat API and API-based BMs in two frequency regions, 3800–2500 cm^−1^ and 1800–400 cm^−1^, are shown in Figure 9. However, firstly, the amorphous FL sample was spectroscopically characterized to recognize/identify the bands arising from hydrogen bonds. Based on the literature data, it is known that FL molecules in the crystalline state are intermolecularly linked by classical N-H···O hydrogen bonds linking one oxygen of the nitro group and the hydrogen of the amide group [79]. Moreover, aromatic hydrogen atoms are involved in C-H···O intermolecular hydrogen bonds with nitro or amide oxygen atoms generating a bifurcated helicoidal hydrogen-bond network. Thus, these functional groups (amide or carbonyl moieties) may have the potential to interact with polymers through the hydrogen bond. As shown in Appendix A, the FTIR spectrum of the crystalline FL confirms the formation of H-bonds by amide groups as the spectral region in the high frequency is dominated by the main peak at 3355 cm^−1^. Several peaks are relatively simple to attribute in this spectrum, such as those at 3130–3000 cm^−1^ (aromatic C–H stretches), 3000–2800 cm^−1^ (aliphatic C-H stretches), 1714 cm^−1^ (C=O stretch), 1611 cm^−1^ (C-C and O-N stretch), 1494 cm^−1^ (HNC deformation), 1313 cm^−1^ (C-C stretch), 1173 cm^−1^ (F-C stretch) and 1139 cm^−1^ (N-C and F-C stretches). A detailed assignment of the IR vibrational bands of crystalline FL was presented in ref. [80]. Compared to the crystalline FL, the ATR-FTIR spectrum obtained for the amorphous API (Appendix A) showed several significant differences, i.e., the narrow peak at 3355 cm^−1^ assigned to the N-H stretching of the amide group is significantly broadened and slightly shifted to lower wavenumbers (two peak maxima at 3309 and 3294 cm^−1^), while the band related to the C=O stretching from the amide group shows a shift from 1714 to 1678 cm^−1^ and consists of two signals of varying intensity. Moreover, there is a change at the 1640–1370 cm^−1^ spectral region of glassy FL, in which some bands exhibit smaller intensity (1617, 1545, 1497 and 1387 cm^−1^) while other new ones appear (1520 and 1416 cm^−1^). Generally, the broadening of most IR bands is associated with a disorder and re-organization of the supramolecular structure of the amorphous network of neat API. In the case of amorphous FL-based systems containing different PVPs, the changes in the profile of N-H stretching band are detected. As can be observed in Figure 9 and Appendix A, the N-H band in the IR spectrum of neat amorphous FL has two maxima at 3309 and 3294 cm^−1^, while the shoulder at higher frequencies partially disappears in FL+PVP mixtures. Additionally, a subtle difference between FL and binary systems was observed in the region of 3185–3025 cm^−1^ assigned to aromatic C-H stretching vibrations (Figure 9). It should be observed that the broadness of this band for FL+10 wt% PVP (M_n_ = 90 kg/mol) mixture appears to be larger than those for the other BMs. On the other hand, the same system is characterized by the lowest intensity of the N–H band shoulder at 3309 cm^−1^. Thus, spectroscopic results reveal that PVP M_n_ = 90 kg/mol may have the greatest effect on the association of drug molecules compared to the other polymers. Similar conclusions from IR spectra analyses were reported in ref. [38] for the FL+PVP K90 system. Moreover, BMs of FL and various PVPs show a higher intensity of the C=O band occurring at 1740–1700 cm^−1^ than the neat API. These little spectral differences between pharmaceutical and solid dispersions, only concerning the bands involved in the formation of H-bonds between FL molecules, may suggest a disturbance of the association of API molecules in the environment of polymers. On the other hand, there are no significant differences in the position of the bands originating from FL molecules in the spectra of the mixtures, and there are no new peaks indicating the existence of intermolecular interactions between the components of the BM. At this point, it should be mentioned that, in the analyzed spectral regions, there is no overlapping the bands originating from PVP (see Appendix A), so the observed spectral changes are only related to the impact of a polymer additive on hydrogen bonding in the amorphous API.

## 3. Materials and Methods

### 3.1. Materials

Flutamide (molecular weight, M_w_ = 276.2 g/mol, purity greater than 99%) and commercially available poly(*N*-vinylpyrrolidone), PVP K90 (PVPcomm., M_w_ = 340 kg/mol, average molecular weight, M_n_~109 kg/mol), were purchased from Sigma-Aldrich and used as received. Two linear PVPs with different M_n_ (=90 kg/mol and 190 kg/mol) as well as hydroxyl-terminated PVP (PVP-OH) with M_n_ = 190 kg/mol were synthesized for the purpose of this paper (for details concerning the synthesis procedure, see the Appendix A and References [33,81].

### 3.2. Preparation of Binary Mixtures

The FL-polymer amorphous binary systems were prepared at different weight concentrations of PVP (PVP-OH) in each sample (10, 30, 50 and 70 wt% of the excipient). To obtain homogeneous samples, we mixed FL with a given PVP (PVP-OH) polymer at appropriate ratios in mortar for 20–30 min. Next, the obtained mixtures were melted at *T*~390 K during the first DSC scan and vitrified by cooling at a rate of 10 K/min during the second DSC scan (standard calorimetric measurements). It should be noted that preliminary miscibility tests and DSC investigations confirmed that only mixtures with 10 and 30% of the polymer are homogeneous. In the systems with a higher content of EXC (>50%), both components, API and polymer, did not mix with each other (the phase separation was noticeable).

Sample preparations for BDS and FTIR studies (only systems with 10 and 30 wt% of the excipient) involved melting at *T*~390 K followed by vitrification on a previously chilled copper plate. All measurements were performed immediately after the preparation of the amorphous BMs to avoid recrystallization.

### 3.3. Differential Scanning Calorimetry (DSC)

Calorimetric measurements of FL, various PVPs (including PVP-OH) and their BMs were performed using a Mettler-Toledo DSC apparatus. This device was equipped with a liquid nitrogen cooling accessory and an HSS8 ceramic sensor having 120 thermocouples. The instrument was calibrated for temperature and enthalpy using indium and zinc standards. The samples were placed in an aluminum crucible (40 μL). Measurements were performed in a temperature range from 260 K to 490 K at a constant heating/cooling rate of 10 K min^−1^. 

Non-isothermal calorimetric measurements (heating rates from 5 to 30 K/min) were carried out on FL+10 wt% PVP (and PVP-OH) BMs immediately after the preparation of amorphous samples. For each experiment, a fresh sample was prepared.

### 3.4. Broadband Dielectric Spectroscopy (BDS)

The dielectric measurements of FL-based BMs were carried out using a Novo-Control GMBH Alpha dielectric spectrometer (Novocontrol Technologies GmbH & Co. KG, Hundsangen, Germany) in the frequency range from 10^−2^ Hz to 10^6^ Hz. The temperature stability controlled by the Quatro System using a nitrogen gas cryostat was better than 0.1 K. Dielectric studies of FL and FL+10 wt% PVP (and PVP-OH) systems were performed immediately after their vitrification in a parallel-plate cell made of stainless steel (diameter 20 mm and 0.1 mm gap with a quartz spacer). Measurements were carried out in a temperature range of 253–317 K.

### 3.5. Fourier Transform Infrared (FTIR) Spectroscopy

The infrared absorption spectra were recorded on a Nicolet™ iS50 Fourier Transform Infrared (FTIR) spectrometer equipped with the built-in diamond Attenuated Total Reflection (ATR) accessory. Each spectrum was the average of 32 scans measured with a spectral resolution of 4 cm^−1^.

### 3.6. Nuclear Magnetic Resonance (NMR) Spectroscopy

^1^H and ^13^C NMR spectra were collected on a Bruker Ascend 500 MHz spectrometer for the samples in CDCl_3_ or D_2_O with a TMS internal standard at 25 °C.

### 3.7. Size Exclusion Chromatography (SEC)

Molecular weights (M_n_) and dispersities (Ð) of PVPs were determined by gel permeation chromatography (GPC) with a Viscotec GPC Max VR 2001 and a Viscotec TDA 305 triple detection-containing refractometer, viscosimeter and low-angle laser-light scattering. The OmniSec 5.12 was used for data processing. Two T6000M general mixed columns were used for separation. The measurements were carried out in DMF (+10 mmol LiBr) as the solvent at 50 °C with a flow rate of 0.8 mL/min. For details (SEC traces of different PVPs used as matrices for FL), see Appendix A. Dispersities of PVP K90 (M_n_ = 109 kg/mol), PVP (M_n_ = 190 kg/mol), PVP (M_n_ = 90 kg/mol) and PVP-OH (M_n_ = 190 kg/mol) determined from chromatographic measurements were as follows: Đ = 1.78, Đ = 1.86, Đ = 1.47 and Đ = 1.63, respectively.

## 4. Conclusions

In this paper, we examined amorphous BMs composed of FL and various PVPs (commercial PVP K90 and synthesized samples) with the use of DSC, BDS and FTIR methods. Calorimetric measurements revealed that in contrast to FL+PVPcomm. solid dispersion, for other FL-based mixtures, there is a large variation between the *T*_g_ of the system with 10 and 30 wt% of the excipient. It was suggested that, most likely, the differences in the tacticity of the polymer forming amorphous BMs are responsible for that. Interestingly, such variations in the tacticity did not significantly affect the *T*_g_ of the studied PVPs themselves, which was in contrast to the literature data reported so far for several polymers. DSC investigations also showed that at higher (i.e., 30 wt%) contents of two new-synthesized polymers, contrary to PVPcomm., the crystallization of FL from the amorphous state is suppressed. Moreover, these studies, together with BDS ones, indicated that, in the mixture with 10 wt% of PVPcomm., FL starts to crystallize at a similar *T* to that of the neat API system. In turn, this process begins at a higher *T* in BMs with other PVPs. Further non-isothermal calorimetric measurements carried out on various FL+10 wt% PVP systems demonstrated that a small amount of each polymer (the most PVP M_n_ = 190 kg/mol) causes the slow down of FL crystallization. It was particularly visible at higher heating rates, i.e., 20 and 30 K/min. The analysis of the collected thermograms with the use of Augis–Bennett and Kissinger approaches revealed that the presence of both commercial, as well as new-synthesized PVPs, only slightly affects the activation energy of non-isothermal crystallization. Finally, infrared investigations demonstrated some small spectral differences, however, only in the bands involved in the formation of H-bonds between API molecules in the examined FL+10 wt% PVP mixtures (the most visible in the case of the BM with PVP, M_n_ = 90 kg/mol) compared to a neat API system. Based on these results, it was suggested that there is a disturbance of the association of API molecules in the environment of polymers. The studies presented herein clearly emphasized that the microstructure of the polymer seems to be a very important parameter that, aside from the molecular weight or dispersity, may contribute to the physical stability of the active substance.

## Figures and Tables

**Figure 1 pharmaceuticals-15-00971-f001:**
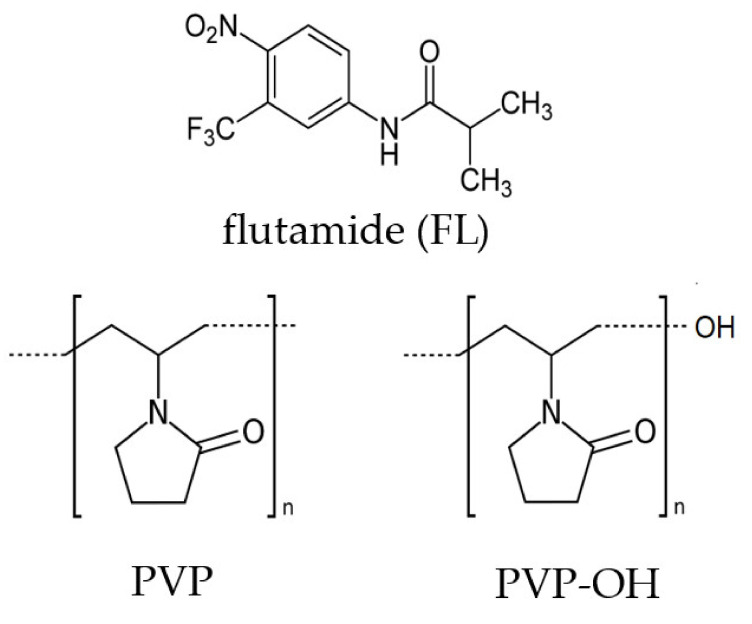
Chemical structures of flutamide (FL) and two polymers: poly-*N*-vinylpyrrolidone (PVP) and hydroxyl-terminated PVP (PVP-OH).

**Figure 2 pharmaceuticals-15-00971-f002:**
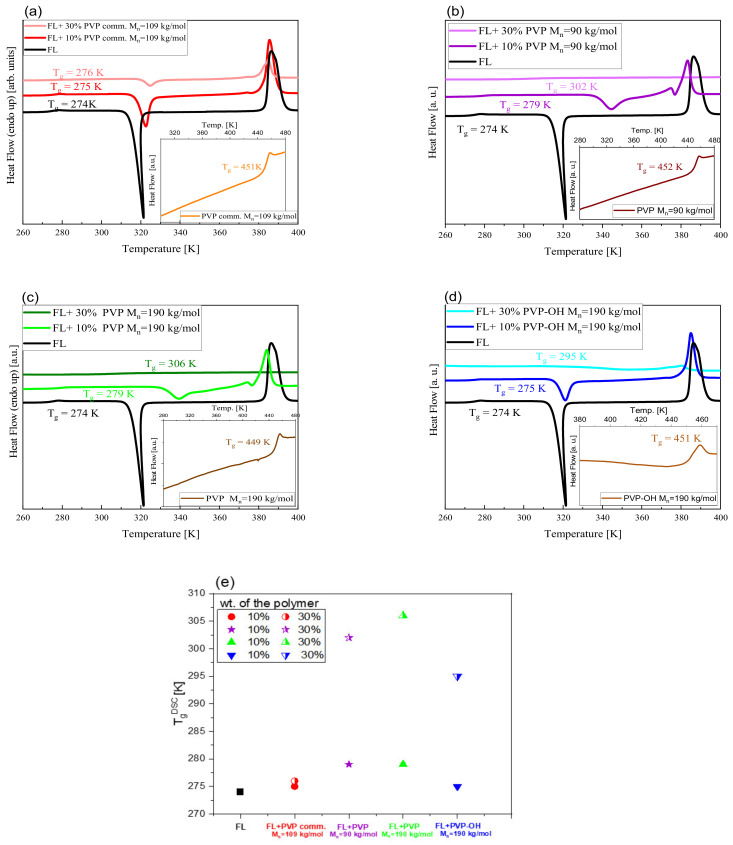
DSC thermograms collected for neat FL and FL-polymer BMs with 10% and 30% content of the excipient (panels (**a**–**d**)). In the insets, DSC curves of neat PVPcomm, PVP M_n_ = 190 kg/mol, PVP-OH M_n_ = 190 kg/mol and PVP M_n_ = 90 kg/mol are also shown. Panel (**e**) presents the glass transition temperatures (*T*_g_^DSC^) of the examined binary systems (with 10 and 30% polymer content). A black square denotes *T*_g_^DSC^ of neat FL.

**Figure 3 pharmaceuticals-15-00971-f003:**
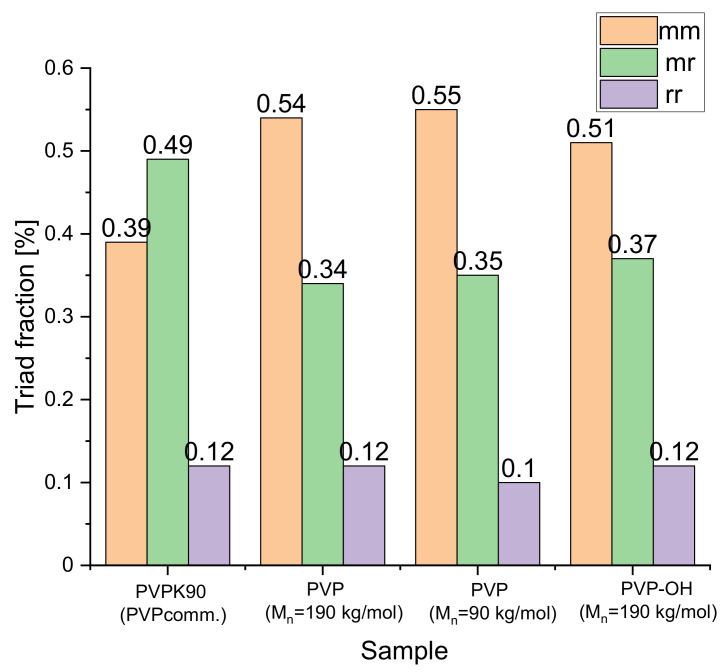
The content of triad fractions determined from *β*-methylene protons (13C NMR, D2O); mm—isotactic triad; mr—atactic triad; rr—syndiotactic triad.

**Figure 4 pharmaceuticals-15-00971-f004:**
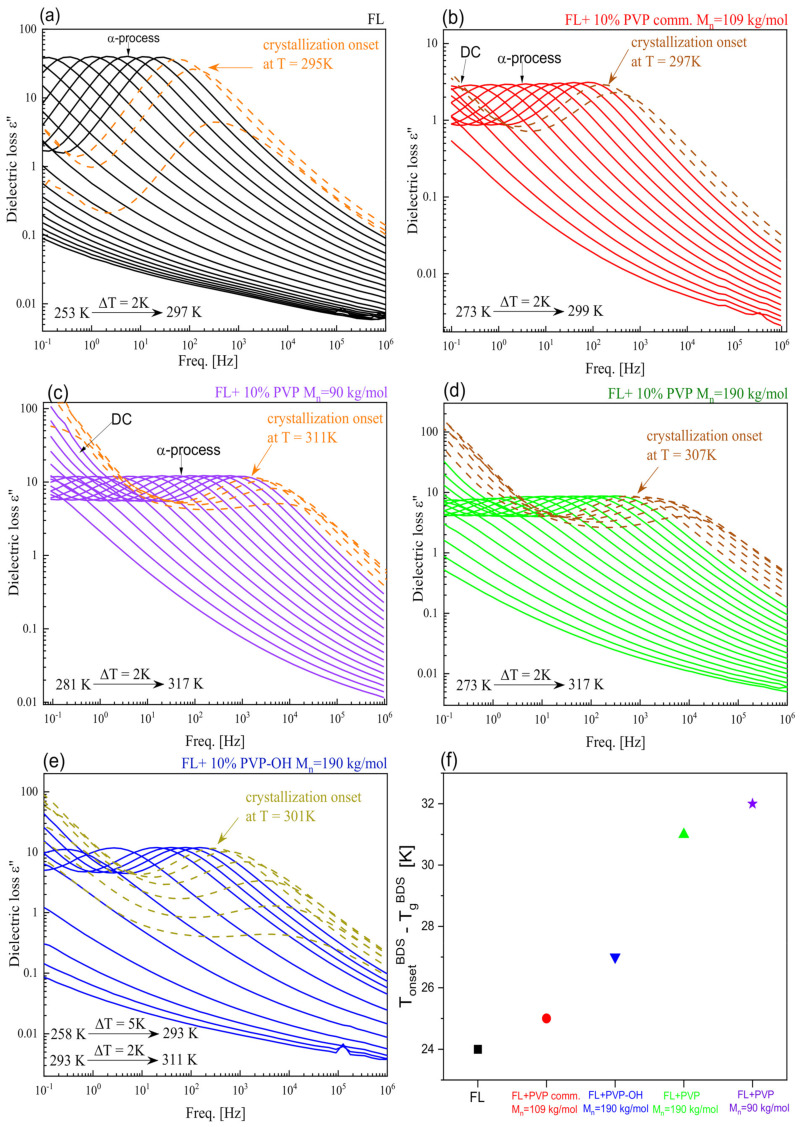
Representative dielectric loss spectra measured for neat FL (**a**) as well as FL+10 wt% PVP(PVP-OH) BMs at *T* > *T*_g_ (**b**–**e**). Panel (**f**) presents the differences between *T*_onset_ and *T*_g_ for FL (a black square) and examined binary systems with 10% content of the polymer: FL+PVPcomm. (a red circle), FL+PVP-OH M_n_ = 190 kg/mol (a blue inverse triangle), FL+PVP M_n_ = 190 kg/mol (a green triangle), FL+PVP M_n_ = 90 kg/mol (a violet star).

**Figure 5 pharmaceuticals-15-00971-f005:**
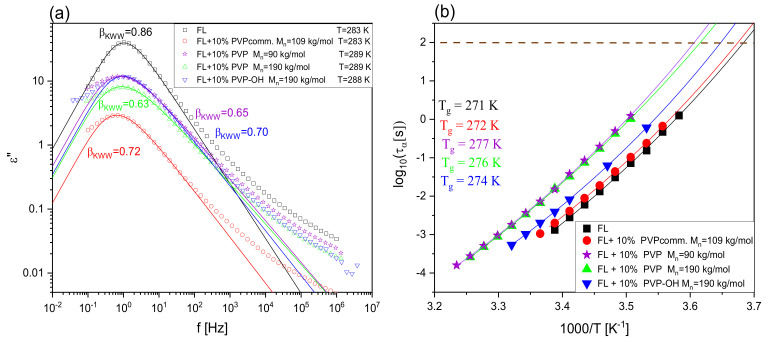
A comparison of the shape of the structural (*α*)-relaxation peak for FL and FL+polymer mixtures with 10 wt% of the latter component at *T*~*T*_g_ (**a**). The solid lines are KWW fits to the data. Temperature dependence of (α)-relaxation times for FL and examined BMs (**b**). The solid lines represent VFT fits.

**Figure 6 pharmaceuticals-15-00971-f006:**
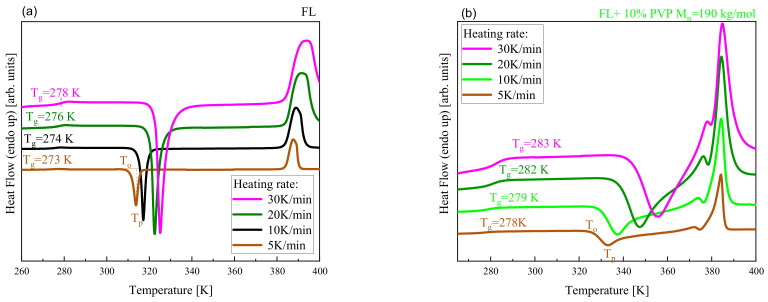
DSC curves obtained for FL (**a**) and its mixture with 10 wt% of PVP M_n_ = 190 kg/mol (**b**). Thermograms were measured with the indicated heating rates.

**Figure 7 pharmaceuticals-15-00971-f007:**
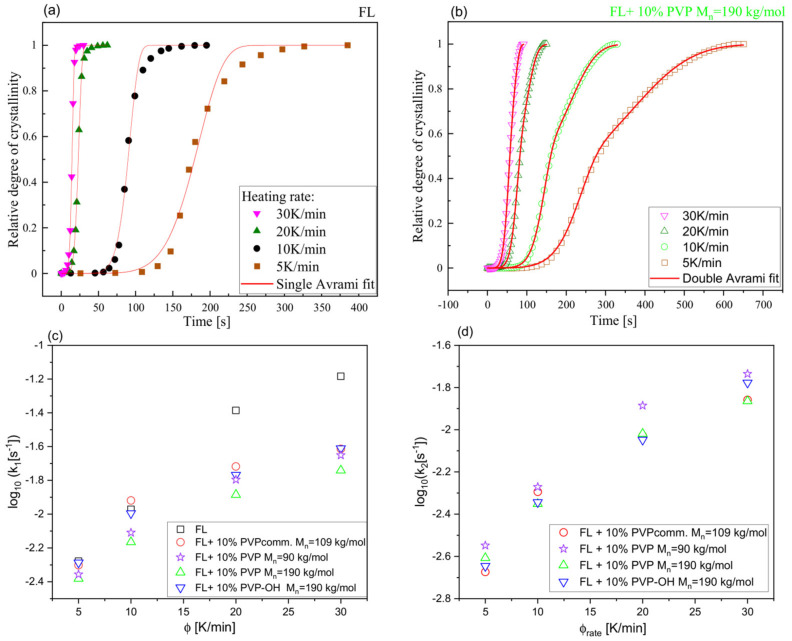
The relative degree of crystallinity (*X*) versus time for FL (**a**) and its mixture with 10 wt% of PVP M_n_ = 190 kg/mol (**b**). In panels (**c**,**d**), the dependences of the logarithm of the crystallization constant rates ($k_{1}$ and $k_{2}$, respectively) versus heating rate are shown.

**Figure 8 pharmaceuticals-15-00971-f008:**
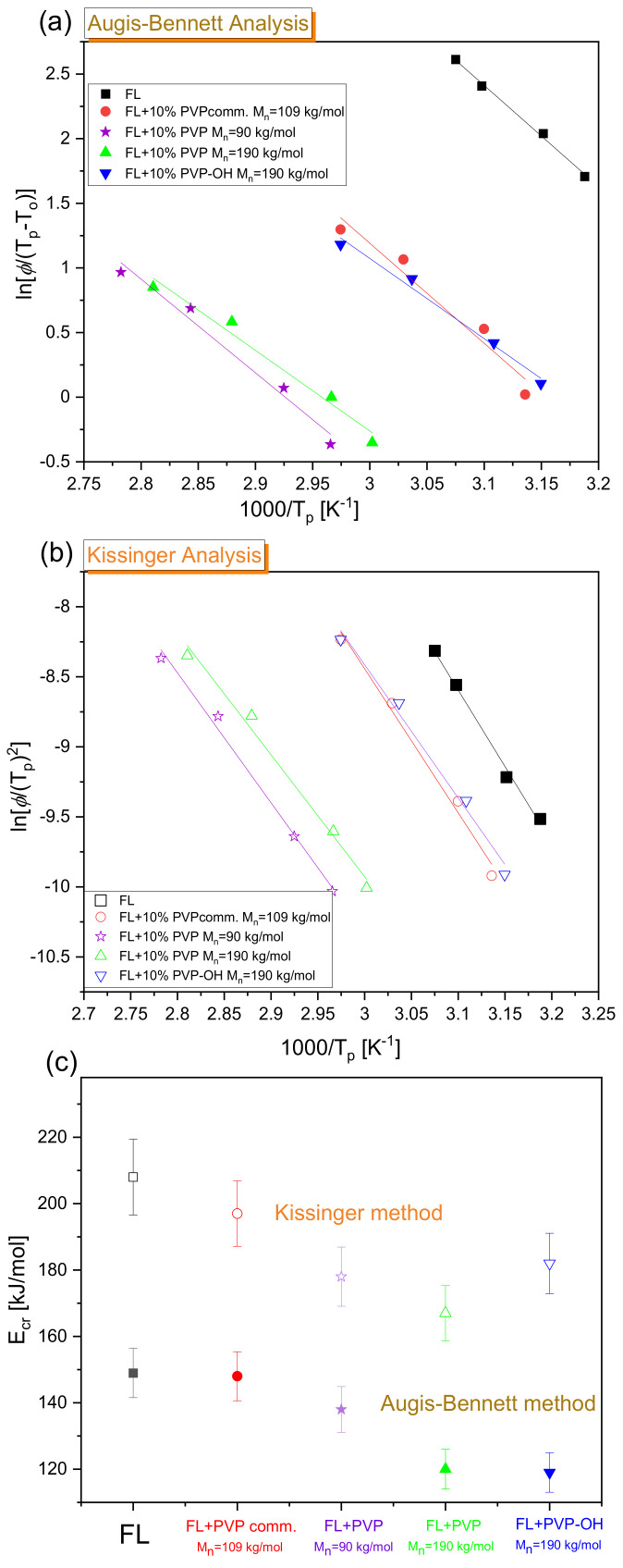
Augis and Bennett (**a**) and Kissinger (**b**) plots for exothermic crystallization peaks in neat FL, FL+PVPcomm., FL+PVP M_n_ = 90 kg/mol, FL+PVP M_n_ = 190 kg/mol and FL+PVP-OH binary mixtures. In panel (**c**), the values of crystallization energy (*E*_cr_) determined for the examined systems from both approaches, along with the error bars (5%), are shown.

**Figure 9 pharmaceuticals-15-00971-f009:**
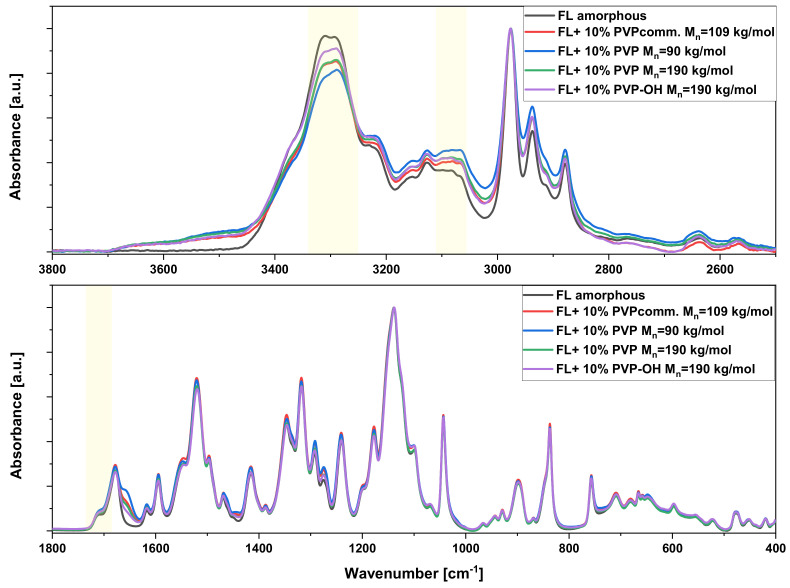
ATR-FTIR spectra of neat FL and its binary mixtures with different PVP and PVP-OH (with 10% polymer content) presented in the high- and low-frequency ranges (3800–2500 cm^−1^ and 1800–400 cm^−1^, respectively). The spectra were normalized to the maximum intensity of the C-H stretching vibration (2976 cm^−1^) and the N-C/F-C stretching vibrations (1139 cm^−1^).

**Table 1 pharmaceuticals-15-00971-t001:** Values of *T*_g_ obtained from BDS measurements as well as *T*_g_ and temperatures of the crystallization process (*T_p_*, *T_o_*) obtained from DSC thermograms collected at different heating rates (ϕ ). Values of the rate constant (*k*_1_, *k*_2_) and Avrami exponent (*n*) obtained the fitting the data presented in Figure 7a (FL), Figure 7b and Appendix A (FL+various PVP binary mixtures) using Equations (6) and (7), respectively, are also given.

System	ϕ (K/min)	*T*_g_ ^BDS^ (K)	*T*_g_ ^DSC^ (K)	$T_{p}\;(K)$	$T_{o}\;(K)$	*k*_1_ (s^−1^)	*k*_2_ (s^−1^)	*n*
FL	5	---	273	313.7	312.1	0.00527	---	1.3
	10	271	274	317.3	316.2	0.01068	---	1.7
	20	---	276	322.8	321.1	0.04113	---	2.2
	30	---	278	325.2	322.9	0.06558	---	2.6
FL+10 wt% PVPcomm M_n_ = 109 kg/mol	5	---	274	318.9	313.9	0.00499	0.00212	2.3
	10	272	275	322.6	318.1	0.01206	0.00508	2.4
	20	---	277	330.1	324.1	0.01916	0.00937	2.6
	30	---	279	336.2	327.9	0.02433	0.01385	2.9
FL+10 wt% PVP M_n_ = 90 kg/mol	5	---	277	337.2	330.2	0.00440	0.00283	2.8
	10	277	279	341.9	334.1	0.00777	0.00535	2.9
	20	---	281	351.7	341.9	0.01601	0.01301	3.0
	30	---	283	359.4	348.1	0.02232	0.01841	2.9
FL+10 wt% PVP M_n_ = 190 kg/mol	5	---	278	333.1	326.2	0.00415	0.00247	3.0
	10	276	279	337.1	330.3	0.00683	0.00445	2.9
	20	---	282	347.3	338.1	0.01302	0.00957	2.6
	30	---	283	355.8	342.9	0.01814	0.01365	2.9
FL+10 wt% PVP-OH M_n_ = 190 kg/mol	5	---	273	317.5	313.1	0.00517	0.00226	2.2
	10	274	275	321.7	317.2	0.01010	0.00453	2.4
	20	---	277	329.3	323.1	0.01705	0.00895	2.8
	30	---	279	336.2	327.2	0.02449	0.01671	2.7

## Data Availability

Data is contained within the article and Appendix A.

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
