# Peer review of "The Effect of Various Poly (N-vinylpyrrolidone) (PVP) Polymers on the Crystallization of Flutamide"

_pharmaceuticals, 2022, doi:10.3390/ph15080971_

Round 1

Reviewer 1 Report

In this manuscript, the authors have studied the effect of mixing various synthetic and common PVA polymers in binary mode on the crystallization of flutamide (FL). The results are compared for synthesized PVA and its mixtures with FL. Depending on the concentration and type of PVA, suppression of crystallization of FL is detected in some cases and the result. The results are exciting and essential to pharmaceutical application, as the crystallization of pharma products has a significant effect on their solubility, stability, and shelf life. However, the manuscript should be modified to make it suitable for publication.

Title: Too long and confusing. I suggest “The effect of poly(N-vinylpyrrolidone) (PVP) polymers on the crystallization of flutamide.”

In the Abstract, change “API” to “active pharmaceutical ingredients” . The term is used before defining it. The Abstract is meant to be self-contained.

Introduction: The interactions of the two polymers are a significant factor in the crystallization of the PVA-FL systems. This is similar to the situation where polymers are loaded with particles to make nanocomposite materials. I think mentioning nanocomposites' crystallization and hybrid polymers in the introduction would widen the relevance to these systems. The effect of adding particles to polymers in retarding crystallization has been studied, and its molecular origins are discussed in the literature, for example, [J. Chem. Phys., 2017, 147, 020901; Nanomaterials, 2019, 9 (10), 1472; Nanoscale Advances., 2019,1, 4704-4721]. Therefore some of the phenomena in retarded crystallization can be explained in these findings in the literature.

“The materials and methods” section should move before the “Results and discussions” section. For example, Scheme 1, which shows the structure of FL , and the two PVA polymers examined in this study, should appear as Figure 1 at the beginning. This makes it much easier to follow the manuscript.

Results:

I suggest the exponent and time constant of the Avrami function be tabulated for various cases and comments made on the effect of PVA addition to these parameters (if any). Some discussion also would be good.

-          Given that the suppression of crystallization is observed for higher concentrations of 30%, I believe the volume fraction effect most likely drives this in hybrid/composite systems. This process has been reported in the literature before (see some references mentioned earlier). What do the authors think about this?

Author Response

Reply to Reviewer 1

We would like to thank the Reviewer for valuable comments that have helped us to significantly improve the manuscript.

In this manuscript, the authors have studied the effect of mixing various synthetic and common PVA polymers in binary mode on the crystallization of flutamide (FL). The results are compared for synthesized PVA and its mixtures with FL. Depending on the concentration and type of PVA, suppression of crystallization of FL is detected in some cases and the result. The results are exciting and essential to pharmaceutical application, as the crystallization of pharma products has a significant effect on their solubility, stability, and shelf life. However, the manuscript should be modified to make it suitable for publication.

1) The Reviewer’s comment:

Title: Too long and confusing. I suggest “The effect of poly(N-vinylpyrrolidone) (PVP) polymers on the crystallization of flutamide.”

Author reply:

Thank you for this comment. The title has been changed according to the Reviewer’s suggestion.

Changes in the manuscript (p. 1):

The effect of various poly(N-vinylpyrrolidone) (PVP) polymers on the crystallization of flutamide

2) The Reviewer’s comment:

In the Abstract, change “API” to “active pharmaceutical ingredients”. The term is used before defining it. The Abstract is meant to be self-contained.

Author reply:

We agree with the Reviewer. Finally, we have decided to use the abbreviation “FL” instead of “API”.

Changes in the manuscript (p. 1):

Further non-isothermal DSC investigations carried out on FL+10 wt% various PVPs mixtures revealed a slowing down of FL crystallization in all FL-based systems (the best inhibitor of this process was PVP Mn=190 kg/mol).

3) The Reviewer’s comment:

Introduction: The interactions of the two polymers are a significant factor in the crystallization of the PVA-FL systems. This is similar to the situation where polymers are loaded with particles to make nanocomposite materials. I think mentioning nanocomposites' crystallization and hybrid polymers in the introduction would widen the relevance to these systems. The effect of adding particles to polymers in retarding crystallization has been studied, and its molecular origins are discussed in the literature, for example, [J. Chem. Phys., 2017, 147, 020901; Nanomaterials, 2019, 9 (10), 1472; Nanoscale Advances., 2019,1, 4704-4721]. Therefore, some of the phenomena in retarded crystallization can be explained in these findings in the literature.

Author reply:

Thank you for this comment. According to it, we have added a part of the text concerning the studies on polymer-based nanocomposites in the Introduction section.

Changes in the manuscript (p. 3):

Herein, one can also mention the volume fraction effect, which is important in the case of hybrid polymer-based nanocomposites. In such systems, the suppressed crystallization of the polymers, as well as self-assembly of the nanoparticles, has been discussed as related to this phenomenon [27,28,29,30].

  1. Kumar, S. K.; Ganesan, V.; Riggleman, R.A. Perspective: Outstanding theoretical questions in polymer-nanoparticle hybrids. J. Chem. Phys. 2017, 147, 020901.

28. Jabbarzadeh, A. The origins of enhanced and retarded crystallization in nanocomposite polymers. Nanomaterials 20199, 1472.

  1. Jabbarzadeh,A.; Halfina,B. Unravelling the effects of size, volume fraction and shape of nanoparticle additives on crystallization of nanocomposite polymers. Nanoscale Adv. 2019, 1, 4704–4721.
  2. Genix, A-C.; Oberdisse, J. Nanoparticle self-assembly: from interactions in suspension to polymer nanocomposites. Soft Matt. 2018, 14, 5161–55179.

4) The Reviewer’s comment:

“The materials and methods” section should move before the “Results and discussions” section. For example, Scheme 1, which shows the structure of FL, and the two PVA polymers examined in this study, should appear as Figure 1 at the beginning. This makes it much easier to follow the manuscript.

Author reply:

According to the journal (Pharmaceutical)’s requirements, the section “Materials and methods” should be after the section “Results and discussion”. For this reason, we have not changed this order. However, according to the Reviewer’s remark, the structures of examined compounds have been presented in a separate figure (Figure 1) at the beginning of the “Results and discussion” section.

Changes in the manuscript (p. 4):

As a first, we have carried out calorimetric measurements on neat FL, various PVPs (including PVP-OH derivative) – see Figure 1, and their mixtures (with 10 and 30 wt% of the second component).

flutamide (FL)

PVP

PVP-OH

Figure 1. Chemical structures of flutamide (FL) and two polymers: poly-N-vinylpyrrolidone (PVP) and hydroxyl-terminated PVP (PVP-OH).

5) The Reviewer’s comment:

I suggest the exponent and time constant of the Avrami function be tabulated for various cases and comments made on the effect of PVA addition to these parameters (if any). Some discussion also would be good.

Author reply:

According to the Reviewer’s comment, we have added additional columns to Table 1 containing the parameters of the Avrami equation (exponent and time constant).

We have also included a brief discussion on these data in the manuscript.

Changes in the manuscript (p. 13,16):

In panels (c) and (d) of Figure 7, dependences of log10  and log10  versus  are presented. As illustrated (Figure 7c, Table 1), a small amount of each polymer (10 wt%) causes the slowing down of FL crystallization. It is especially noticeable at higher heating rates: 20 and 30 K/min (smaller  when compared to that of neat FL). PVP Mn=190 kg/mol is the excipient whose impact on such behavior is most significant (the lowest values of at each  see Table 1). Comparing polymers with the same/nearly the same Mn, it can be stated that PVP Mn=90 kg/mol and PVP Mn=190 kg/mol, compared to PVPcomm. and PVP-OH, respectively, suppress/slow down the API crystallization a little bit more (Figure 7c). In the context of the constant rate one can add that its value determined for all binary systems at a given  was comparable (Table 1). It should also be mentioned that the values of the exponent  obtained from the analysis of the data for neat FL using Eq. (6) and FL+10 wt% PVP (PVP-OH) using Eq. (7) changed generally in the range 2-3 - a somewhat lower n (1.3-1.7) were determined only for neat FL at lower , see Table 1). In turn, the values of the parameter  (Eq. 7) were clearly higher and unreliable. Therefore, we will not present (in Table 1) and discuss them.

Table 1. Values of Tg obtained from BDS measurements as well as Tg and temperatures of the crystallization process (Tp, To) obtained from DSC thermograms collected at different heating rates ( ). Values of the rate constant (k1, k2) and Avami exponent (n) obtained the fitting the data presented in Figure 7a (FL) and Figures 7b and S7 (FL+various PVP binary mixtures) using Eq. (6) and Eq. (7), respectively, are also given.

System

 [K/min]

TgBDS [K]

TgDSC [K]

 [K]

 [K]

k1 [s-1]

k2 [s-1]

n

FL

5

---

273

313.7

312.1

0.00527

---

1.3

10

271

274

317.3

316.2

0.01068

---

1.7

20

---

276

322.8

321.1

0.04113

---

2.2

30

---

278

325.2

322.9

0.06558

---

2.6

FL+10 wt% PVPcomm.

Mn=109 kg/mol

5

---

274

318.9

313.9

0.00499

0.00212

2.3

10

272

275

322.6

318.1

0.01206

0.00508

2.4

20

---

277

330.1

324.1

0.01916

0.00937

2.6

30

---

279

336.2

327.9

0.02433

0.01385

2.9

FL+10 wt% PVP

Mn=90 kg/mol

5

---

277

337.2

330.2

0.00440

0,00777

0,016

0,02232

0.00283

2.8

10

277

279

341.9

334.1

0.00777

0.00535

2.9

20

---

281

351.7

341.9

0.01601

0.01301

3.0

30

---

283

359.4

348.1

0.02232

0.01841

2.9

FL+10 wt% PVP

Mn= 190 kg/mol

5

---

278

333.1

326.2

0.00415

0.00247

3.0

10

276

279

337.1

330.3

0.00683

0.00445

2.9

20

---

282

347.3

338.1

0.01302

0.00957

2.6

30

---

283

355.8

342.9

0.01814

0.01365

2.9

FL+10 wt% PVP-OH

Mn=190 kg/mol

5

---

273

317.5

313.1

0.00517

0.00226

2.2

10

274

275

321.7

317.2

0.01010

0.00453

2.4

20

---

277

329.3

323.1

0.01705

0.00895

2.8

30

---

279

336.2

327.2

0.02449

0.01671

2.7

6) The Reviewer’s comment:

      Given that the suppression of crystallization is observed for higher concentrations of 30%, I believe the volume fraction effect most likely drives this in hybrid/composite systems. This process has been reported in the literature before (see some references mentioned earlier). What do the authors think about this?

Author reply:

Thank you for this comment. Referee is right that similarly to nanocomposite systems, the volume fraction effect can be responsible for the suppression of crystallization in FL-PVPsynt. mixtures with higher (>30%) concentration of the excipient. A brief comment about this issue has been added to the text of the manuscript.

Changes in the manuscript (p. 6):

One can suppose that the slowing down of the crystallization in the case of two examined herein systems (> 30 wt% of PVPsynt. Mn=90 and 190 kg/mol) can be related to the greater volume fraction of polymers. Importantly, a similar phenomenon was argued to be responsible for the suppression of the the crystallization and self-assembly process of nanoparticles in polymer-based nanocomposite materials [28,29,30].

Reviewer 2 Report

In this paper the authors have done DSC, BDS, and FTIR experiments on binary mixtures of flutamide and poly(N-vinylpyrrolidone). Some unexpected results on the glass transition temperature of the polymer are reported and discussed. The main finding is that the microstructure of the polymer is an important parameter, which contributes to the physical stability of the pharmaceuticals. The manuscript is well written and understandable. However, I have two major concerns, to be addressed to, before publication:

1-Page 6, it is discussed that the tacticity does not have a noticeable effect on the glass transition temperature of the polymers studied. This independency is interpreted in terms of the philicity of the polymer. However, experimental reports in the literature on the tacticity-dependence of the glass transition temperature of polar polymers reveals that tacticity considerably influences the glass transition temperature. Perhaps the authors need to better clarify this point.

2-Figure 2 and the text on page 10: The time-dependence of the relative degree of crystallization is discussed in terms of a two-stage crystallization process. However, the results in panel (a) of the figure do not look like a two-stage crystallization process. In a two-stage crystallization process, in the first step dense liquid (disorder) clusters are formed. In the second step the molecules in the dense liquid cluster rearrange their orientations to form ordered nuclei (crystals). See for example (J. Chem. Theory Comput. 2022, 18, 1870 and J. Chem. Theory Comput. 2021, 17, 1742). Please clarify this point. Also I recommend that the authors write a few lines regarding the two-stage crystallization process in the text.

Author Response

Reply to Reviewer 2

We would like to thank the Reviewer for valuable comments that have helped us to significantly improve the manuscript.

In this paper the authors have done DSC, BDS, and FTIR experiments on binary mixtures of flutamide and poly(N-vinylpyrrolidone). Some unexpected results on the glass transition temperature of the polymer are reported and discussed. The main finding is that the microstructure of the polymer is an important parameter, which contributes to the physical stability of the pharmaceuticals. The manuscript is well written and understandable. However, I have two major concerns, to be addressed to, before publication:

1) The Reviewer’s comment:

Page 6, it is discussed that the tacticity does not have a noticeable effect on the glass transition temperature of the polymers studied. This independency is interpreted in terms of the philicity of the polymer. However, experimental reports in the literature on the tacticity-dependence of the glass transition temperature of polar polymers reveals that tacticity considerably influences the glass transition temperature. Perhaps the authors need to better clarify this point.

Author reply:

Thank you for this comment. We agree with the Reviewer that in the case of less or more polar polymers studied in the literature (e.g., poly(methyl methacrylate (PMMA), poly(2-methoxyethylacrylate) PMEA, polystyrene, polypropylene), tacticity considerably influences the glass transition temperature, degree of crystallinity and properties of the polymer. It was mentioned in the initial version of the article (p. 9).

We suggested that the contradictory effect observed for PVP might be related to the highly polar character of PVP and, contrary to the other polymers, its highly hydrophilic nature. One can also suppose that the residual water, which is always present in the structure of this polymer (it is very difficult to remove it even by heating/annealing) can weakly interact via H-bonds with the macromolecule. For this reason, the effect of Tg on tacticity is clearly lower when compared to other mentioned polymers.

The above (additional) explanation has been included in the revised version of the manuscript.

Changes in the manuscript (p. 8):

Interestingly, the mentioned differences in the tacticity do not significantly affect the glass transition temperature of the studied polymers (they are characterized by a similar Tg, in the range 449-452 K). Hence, obtained results stay in contrast to the literature data showing that there are usually much greater variations in the Tg of more or less polar macromolecules (e.g., poly(methyl methacrylate) (PMMA), poly(2-methoxyethylacrylate) (PMEA) [50,51,52,53], polystyrene [54], etc. [55]), differing in tacticity (i.e., with various content of isotactic (i), syndiotactic (s), atactic (a) fractions in the sample). What is more, in the case of polypropylene, variation in the microstructure leads to a change in the degree of crystallinity, and mechanical properties [55]. The observed discrepancies might be related to the highly polar character of PVP and, in contrary to the other polymers, its highly hydrophilic nature. One can also suppose that the residual water, which is always present in the structure of this polymer (it is very difficult to remove it even by heating/annealing) can weakly interact via H-bonds with the molecules of macromolecule. For this reason, the effect of Tg of PVP on tacticity is clearly lower when compared to other mentioned polymers.

2) The Reviewer’s comment:

Figure 2 and the text on page 10: The time-dependence of the relative degree of crystallization is discussed in terms of a two-stage crystallization process. However, the results in panel (a) of the figure do not look like a two-stage crystallization process. In a two-stage crystallization process, in the first step dense liquid (disorder) clusters are formed. In the second step the molecules in the dense liquid cluster rearrange their orientations to form ordered nuclei (crystals). See for example (J. Chem. Theory Comput. 2022, 18, 1870 and J. Chem. Theory Comput. 2021, 17, 1742). Please clarify this point. Also I recommend that the authors write a few lines regarding the two-stage crystallization process in the text.

Author reply:

Thank you for this comment.

As mentioned in the initial version of the paper, the two-step crystallization was considered and analyzed (with the use of the modified Avrami model - Eq. (7) and) only for FL-various PVP 10 wt% binary systems; see Figure 7b and S7. In the case of neat FL, a single Avrami equation (which is commonly used to describe a single-step process) was applied to fit the obtained data (Figure 7a).

According to the Reviewer’s remark, we have added a few lines regarding the two-stage crystallization process in the text.

Changes in the manuscript (p. 13):

It is worth pointing out that according to the literature reports on a two-stage crystallization process (however taking place from solution/fluid, not from the solid state), in the first step dense liquid (disorder) clusters are formed, while in the second step, the molecules in the mentioned clusters rearrange their orientations to form ordered nuclei (crystals). Such mechanism, which was suggested also in the case of e.g., triblock Janus assemblies [65,66,67], proteins [68], colloids [69], and biomaterials [70], is sometimes more favorable than the one-step formation of crystalline nuclei. However, it should be noted that since we measured solid state systems, the two-stage process might be due to the change of one polymorphic form into another during the course of crystallization.

  1. Bahri, K.; Eslami, H.; Müller-Plathe, F. Self-assembly of model triblock Janus colloidal particles in two dimensions. J. Chem. Theory Comput. 2022, 18, 1870–1882.
  2. Eslami, H.; Gharibi, A.; Müller-Plathe, F. Mechanisms of nucleation and solid–solid-phase transitions in triblock Janus assemblies. J. Chem. Theory Comput.2021, 17, 1742–1754.

67. Romano, F.; Sciortino, F. Crystal phases of two dimensional assembly of triblock Janus particles. Soft Matter 2011, 7, 5799−5804.

  1. 68. ten Wolde, P.R.; Frenkel, D. Enhancement of protein crystal nucleation by critical density fluctuations. Science 1997, 277, 1975−
  2. Lee, S.; Teich, E.G.; Engel, M.; Glotzer, S.C. Entropic colloidal crystallization pathways via fluid-fluid transitions and multidimensional prenucleation motifs. Proc. Natl. Acad. Sci. U.S.A. 2019, 116, 14843−14851.
  3. Pouget, E.M.; Bomans, P.H.H.; Goos, J.A.C.M.; Frederik, P.M.; de With, G.; Sommerdijk, N.A.J.M. The initial stages of template-controlled CaCO3 formation revealed by cryo-TEM. Science 2009, 323, 1455−1458.

Round 2

Reviewer 1 Report

I believe the authors have addressed the issues raised in the first round of review, the paper can now be published.

Reviewer 2 Report

The revised version of manuscript is improved considerable. I recommend publication.